# Isoliquiritigenin Derivatives Inhibit RANKL-Induced Osteoclastogenesis by Regulating p38 and NF-κB Activation in RAW 264.7 Cells

**DOI:** 10.3390/molecules25173908

**Published:** 2020-08-27

**Authors:** Seongtae Jeong, Seahyoung Lee, Kundo Kim, Yunmi Lee, Jiyun Lee, Sena Oh, Jung-Won Choi, Sang Woo Kim, Ki-Chul Hwang, Soyeon Lim

**Affiliations:** 1Institute for Bio-Medical Convergence, College of Medicine, Catholic Kwandong University, Gangneung-si, Gangwon-do 210-701, Korea; 91seongtae@gmail.com (S.J.); sam1017@ish.ac.kr (S.L.); jylee12334@gmail.com (J.L.); bole1305@naver.com (S.O.); jungwonjian@gmail.com (J.-W.C.); doctor7408@gmail.com (S.W.K.); 2Department of Chemistry, Kwangwoon University, Seoul 01897, Korea; vmczl_@naver.com (K.K.); ymlee@kw.ac.kr (Y.L.)

**Keywords:** isoliquiritigenin derivatives, robtein, RANKL, osteoclastogenesis

## Abstract

Bone diseases may not be imminently life-threatening or a leading cause of death such as heart diseases or cancers. However, as aging population grows in almost every part of the world, they surely impose significant socioeconomic burden on the society, not to mention the patients and their families. Osteoporosis is the most common type of bone disease, which frequently develops in seniors, especially in postmenopausal women. Although currently several anti-osteoclastic drugs designed to suppress excessive osteoclast activation, a major cause of osteoporosis, are commercially available, accompanying adverse effects ranging from mild to severe have been reported as well. Natural products have become increasingly popular because of their effectiveness with fewer side effects. Isoliquiritigenin (ILG), a natural flavonoid from licorice, has been reported to suppress osteoclast differentiation and activation. In the present study, newly synthesized ILG derivatives were screened for their anti-osteoporotic activity as more potent substitute candidates to ILG. Out of the 12 ILG derivatives tested, two compounds demonstrated significantly improved bone loss in vitro by inhibiting both osteoclastogenesis and osteoclast activity. The results of the present study indicate that these compounds may serve as a potential drug for osteoporosis and warrant further studies to evaluate their in vivo efficacy.

## 1. Introduction

The osteoclast derived from hematopoietic stem cells, a specialized multinucleated cell to resorb bone, and the osteoblast derived from mesenchymal stem cells, a single nucleus cell to synthesize bone, are the major cell types consisting in bones. Within the skeleton, these cells maintain the architecture of bones and facilitate remodeling by secreting various signaling molecules, such as cytokines and chemokines [1]. Therefore, an imbalance between osteoclast and osteoblast activity contributes to the development of several bone diseases. For example, excessive osteoclast activation has been implicated in the occurrence of osteoporosis, periprosthetic osteolysis, rheumatoid arthritis, and subchondral bone destruction [2]. In particular, osteoporosis is the most common type of bone disease due to osteoclast activation worldwide, and it frequently develops in seniors, especially in postmenopausal women. Excessive amounts of reactive oxygen species or inflammatory cytokines are regarded as the main cause of hyperactive osteoclast formation, and monocyte/macrophage colony-stimulating factor (M-CSF) and receptor activator of nuclear factor kappa-B (NF-κB) ligand (RANKL) are known as the two essential factors for the hyperactive osteoclast formation since 1990s [3,4]. M-CSF plays roles in proliferation and survival of osteoclast precursors and RANKL functions in the differentiation of osteoclast precursors into mature osteoclasts. RANKL, mainly expressed on osteoblasts, stromal cells, and T cells, is secreted into the extracellular environment and directly binds to RANK expressed on osteoclast precursors and mature osteoclasts [5,6]. RANKL binding to RANK leads to the activation of downstream signaling mediators, namely mitogen-activated protein kinases (MAPKs), nuclear factor-κB (NF-κB), and activator protein-1 (AP-1), by recruiting adaptor molecules such as TNFR-associated factors (TRAFs) [7]. AP-1 and NF-κB then induce nuclear factor of activated T-cells cytoplasmic 1 (NFATc1) transcription [8]. In addition to RANK-mediated activation, it is known that immunoglobulin-like receptors such as triggering receptor expressed in myeloid cells-2 (TREM-2) and osteoclast-associated receptor (OSCAR) cooperate with RANK-mediated signaling to activate NFATc1 [9]. Activated NFATc1 can increase the expression of osteoclast-specific genes such as tartrate-resistant acid phosphatase (TRAP), β_3_-integrin, dendritic cell-specific transmembrane protein (DC-STAMP), cathepsin K (CTSK), and calcitonin receptors (Calcr), that act on various stages of osteoclast development and activation [10]. Over 24 genes are known to be involved in the regulation of osteoclastogenesis and osteoclast activation [11].

To date, various medications for osteoporosis have been developed and are available for clinical treatment. Although agents such as estrogen, selective estrogen receptor modulator (SERM), bisphosphates, and calcitonin, a major anti-resorptive drug, have made remarkable progress for both prevention and treatment [12,13,14,15], severe adverse effects remain as a hurdle in clinical management of osteoporosis. It has been reported that a long-term treatment of estrogen may increase cardiovascular events and breast cancer risk in the Women’s Health Initiative (WHI) study [16]. A long-term use of raloxifene, a kind of SERMs, and calcitonin was reported to increase the incidences of venous thromboembolism and risk of cancer, respectively [17,18]. Furthermore, Bisphophonates, an effective drug for osteoporosis [19,20], can also cause rare side-effects, such as femur fractures and osteonecrosis of the jaw (Figure 1).

Although the number of new anti-resorptive drugs such as sclerostin inhibitors and cathepsin K inhibitors is increasing, many patients are not taking them because of rare, but severe side effects. Thus, it is necessary to develop alternatives with minimized adverse effects.

Natural products have become increasingly popular because of their effectiveness with fewer side effects. Among them, isoliquiritigenin (ILG), a natural flavonoid from licorice, has been reported to suppress osteoclast differentiation [21,22,23]. Additionally, the beneficial effects of ILG in various disease models such as diabetes, cancers, and obesity are due to its anti-oxidant, anti-inflammation, or anti-proliferation/migration effect [24,25,26].

In the present study, 12 ILG derivatives were synthesized and screened for their activities against osteoclast differentiation. Our in vitro data indicated that two compounds out of 12 that had been tested showed potential therapeutic effects by in vitro experiments.

## 2. Results

### 2.1. ILG Derivatives Inhibit RANKL-Induced Osteoclast Differentiation

To investigate whether ILG derivatives can effectively inhibit osteoclast differentiation induced by RANKL, ILG and its 12 derivatives were synthesized (Scheme 1, Appendix A). To induce osteoclastogenesis, RAW264.7 cells, commonly used as osteoclast precursors derived from murine leukemia virus-induced tumor, were incubated with 40 ng/mL RANKL with or without various concentration (0.1–10 μM) of ILG or 12 different ILG derivatives. RANKL-induced osteoclast differentiation was evaluated by TRAP staining (Figure 2A and Appendix A), in which two out of total 12 derivatives exhibited improved anti-osteoclastogenesis potential compared to that of ILG (10 μM) (Figure 2B). Moreover, derivative three and ten significantly decreased the number of TRAP^+^-osteoclast (nuclei ≥ 3) compared to ILG in a dose-dependent manner (Figure 2C). For RANKL-induced increase of TRAP mRNA expression, although all the derivatives tended to suppress the increase of the RANKL-induced TRAP mRNA expression, it was not statistically significant, except the derivative 10 (Figure 2D,E). The cell viability (Figure 3A) and cytotoxicity (Figure 3B) were also investigated by cell viability assay and lactate dehydrogenase (LDH) activity assay, respectively. Increasing concentrations (0, 1, 5, 10 μM) of the derivative 3 and 10 did not cause any significant changes of viability or cytotoxicity. These results suggested that the derivative 3 and 10 may have a better potential in inhibiting the RANKL-induced osteoclast differentiation than ILG.

### 2.2. ILG Derivatives Impaired Bone Resorptive Activity of RANKL-Induced Osteoclasts

To further examine the anti-osteoclastogenic activity of ILG derivatives, expressions of osteoclast-specific genes (Calcr, CTSK, and DC-STAMP), that are known to be predominantly expressed in active osteoclasts, were investigated. For example, a collagenase CTSK is known to degrade bone mineral and collagen matrices, and DC-STAMP plays a critical role for osteoclast cell fusion [10,27]. According to the data, ILG, the derivative 3, and 10, significantly suppressed the RANKL-induced upregulation of mRNA expressions of these genes in a concentration-dependent manner (Figure 4). The active osteoclasts show a resorption activity which can lead to pit formation characterized by clusters of round excavation [28]. For the assessment of osteoclast functions, such as bone resorption, F-actin ring formation and pit formation assays were performed [29]. RANKL-induced F-actin ring formation was apparently suppressed by ILG, the derivative 3, and 10, and higher concentration (10 μM) showed a more pronounced reduction of F-actin ring formation compared to lower concentration (5 μM) for all three compounds (Figure 5A). Consistent with the results on F-actin ring formation, ILG, the derivative 3, and 10 also significantly suppressed the RANKL-induced osteoclast bone resorption (Figure 5B,C). These data clearly demonstrated that ILG, as well as its derivatives 3 and 10, could significantly inhibit the RANKL-induced bone resorbing function and osteoclast formation.

### 2.3. Combinational Treatment of ILG Derivatives Showed a Synergistic Effect on RANKL-Induced Osteoclast Differentiation and Activation

To examine any possible synergistic effect of the derivative 3 and 10, the effect of combinational treatment of the derivative 3 and 10 was evaluated. The results of TRAP staining assay indicated that the combination of the derivative 3 and 10 was more effective than single individual treatment (Figure 6A,B), without any significant cytotoxicity (Figure 6C). For the functional activity of combinational treatment, the reduction of pit areas were similar to that of derivative 10 (5 μM) (Figure 6D).

### 2.4. ILG Derivatives Suppressed RANKL-Induced Osteoclast Differentiation Signaling Pathways

To elucidate the mechanism with which ILG, the derivative 3, and the derivative 10 inhibit the RANKL-induced osteoclast differentiation, their effects on MAPKs and NF-κB signaling pathways, well-known main signaling pathways for osteoclast differentiation and activation [30], were investigated. According to the data, RANKL significantly induced phosphorylation of MAPKs (p38, ERK1/2, and JNK) and NF-κB. Among these signaling pathways, ILG, the derivative 3, and the derivative 10 selectively attenuated the activation of p38 and NF-κB (Figure 7). These data indicated that even a lower concentration of the derivative 3 or 10 can effectively inhibit osteoclast differentiation and activation than ILG treatment by downregulating p38 and NF-κB pathways.

## 3. Discussion

ILG has been reported to inhibit RANKL-induced osteoclast differentiation [21]. In the present study, we have synthesized 12 additional derivatives of ILG, which varied the number and positions of amino, nitro, trifluoromethyl, methoxy, hydroxy and halogen substituents on the two phenyl rings, and examined their possible effect on the RANKL-induced osteoclast differentiation. Our results demonstrated that both ILG and its derivatives 3 and 10 showed effective inhibition against RANKL-induced osteoclast differentiation. These three compounds shared common regulatory mechanism to control osteoclast differentiation (Figure 7), and they showed concentration-dependent attenuation of the increases in the number of TRAP^+^-osteoclast (nuclei ≥ 3), osteoclast-specific gene expressions, and pit areas at the concentration of 10 μM. In general, a lower concentration of compound can minimize the incidence of adverse drug effects [31]. Therefore, combinations of these compounds with lower concentrations were tested for their effects on osteoclast differentiation to see if there was any combination with possibly minimized adverse effect. According to the TRAP staining data (Figure 6A), a combination of lower dose derivative 3 (1 μM) and derivative 10 (5 μM) showed more protective effect than individual treatment with derivative 3 (5 μM) or derivative 10 (5 μM). However, other related data such as bone resorption, TRAP^+^-osteoclast (*n* ≥ 3), and Western blotting failed to show a significant advantage of using combinatorial treatment. Although there still may exist an optimized regiment for an effective combinatorial treatment, at this point, without any further evidence to prove or disapprove such possibility, the ILG-derived derivative 10 is the most effective treatment for preventing osteoclast differentiation and activation.

Derivative 10, known as robtein, belongs to a chalcone class previously called an anthochlor pigment [32]. Unlike ILG, little has been known for the biological effects of robtein. However, certain characteristics of robtein may be speculated as a chalcone family compound. Previous studies have demonstrated that the chalcone and its derivatives possess anti-inflammatory, anti-oxidant, and anti-cancer functions, suggesting a pharmacological potential of the chalcones [33,34,35]. Moreover, another beneficial effect such as estrogenic effect of robtein as a chalcone compound can be speculated. ILG, as a primary bioactive chalcone compound in licorice root, demonstrated estrogenic activity both in vivo and in vitro. Maggiolini et al. showed that low levels of ILG had estrogenic and anti-proliferative activities in breast cancer cells, and Mersereau et al. demonstrated that ILG is a selective estrogen receptor beta agonist [36,37]. Butein, another member of chalcone family, was also reported to modulate estrogen metabolism [38]. Therefore, although we did not experimentally cover the subject in the present study, it is highly possible that ILG derivatives 3 and 10 also possess estrogenic effect based on the reported characteristics of chalcones [39].

ILG itself has been continuously suggested as a dietary supplement or alternative therapeutic agent for several cancers, and clinical studies have not reported any significant side effects to date [35]. Although ILG has been reported to have an anti-proliferative effect in cancer models at 25–50 μM [35,40], according to our data, both the derivative 3 and 10 only caused insignificant reduction of cell viability under 10 μM (Figure 3), indicating that the lower concentration of robtein may be acceptable for biological effects without causing any significant adverse effect.

In the investigation of a compound for possible pharmacological effects, timing of treatment is important. According to the previous study, ILG acts on the initial steps of RANKL-induced osteoclast differentiation. To be specific, ILG showed inhibitory effect on osteoclast differentiation only when co-treatment with RANKL, but it failed to reverse the F-actin ring formation by post-treatment [21]. Similar to such observation, ILG derivatives examined in the present study also showed a significant inhibitory effect on the RANKL-induced osteoclast differentiation and activation when they were applied prior to RANKL induction. Therefore, although additional study may be required to validate, it can be speculated that ILG derivatives are useful as a preventive and/or protective therapeutic agent rather than as a treatment option. In this sense, ILG derivatives can be a good candidate for a major ingredient of dietary supplement or neoadjuvant agent targeting osteoporosis.

Lastly, there are some limitations of this study to be pointed out. In the present study, we performed pit assays and analyzed cathepsin K expression to confirm the regulatory role of ILG derivatives against osteoclast activation. However, calcium phosphate-coated plate which we used, represents inorganic component of bones, not organic component such as collagen [41]. In addition, estimation of expression levels of carbonic anhydrase II and V-ATPase, besides of cathepsin K would be better to explain the regulatory role of ILG and its derivatives, because these two factors have essential roles in bone resorption by increasing acid (H^+^) secretion [42,43].

The present study is the first report to demonstrate that robtein, a derivative of ILG, has a potential to be an effective anti-osteoclastic drug and calls for further in vitro studies to validate its therapeutic efficacy as well as of its derivatives.

## 4. Materials and Methods

### 4.1. Synthesis

ILG and its 12 derivatives were prepared through Claisen–Schmidt condensation of acetophenones with benzaldehydes, followed by deprotection under acidic conditions, and general methods are described in Scheme 1.

### 4.2. Cell Lines

Mouse monocyte macrophage derived from Abelson murine leukemia virus-induced tumor, Raw264.7 cell (TIB-71; ATCC, Manassas, VA, USA), was purchased from American Type Culture Collection (ATCC). Raw264.7 cells were maintained in Dulbecco’s modified Eagle’s medium (DMEM) (30-2002; ATCC) with 10% FBS (16000-044, Thermo Fisher Scientific, Waltham, MA, USA), 100 U/mL penicillin (15140-122, Thermo Fisher Scientific), and 100 μg/mL streptomycin (15140-122; Thermo Fisher Scientific). Raw264.7 cells were incubated at 37 °C in 5% CO_2_ humidified atmosphere. Raw264.7 cells were used between passage 5 and 20. For cell differentiation and drug treatment, Raw264.7 cells were cultured in complete alpha modified minimal essential medium (α-MEM) (11900-024, Thermo Fisher Scientific).

### 4.3. Cell Viability and Toxicity

To detect cell viability and toxicity of the compounds, we conducted cell viability assay (EZ-cytox; Dogenbio, Seoul, Korea) and lactate dehydrogenase (LDH) activity assay (LDH Cytotoxicity Detection kit; TAKARA, Otsu, Japan) under the manufacturer’s protocols. Raw264.7 cells were seeded in 96-well culture plate at the density of 2.5 × 10^3^ cells per well and allowed to adhere for 24 h. Raw264.7 cells were treated with DMSO (vehicle), different concentration of ILG and ILG derivatives (1, 5, and 10 μM) in α-MEM medium for 48 h at 37 °C in 5% CO_2_. At 48 h after treatment, to detect cell viability, 10 μL of EZ-cytox solution was added to all wells and incubated at 37 °C for 2 h. Absorbance of formazan concentrations was measured at 450 nm using a microplate reader (Multiskan FC; Thermo Fisher Scientific). To measure the cell toxicity of ILG and ILG derivatives, we collected cell supernatants after treatment and centrifuged at 250× *g* for 10 min. Total of 100 μL of LDH reagent was added into 100 μL of supernatants and incubated at room temperature for 30 min. The activity of LDH in the supernatant was detected by microplate reader at a wavelength of 490 nm. The obtained absorbance values were used to calculate percentage of vehicle.

### 4.4. TRAP Staining

The osteoclastogenesis was measured by quantifying a tartrate-resistant acid phosphatase (TRAP) of osteoclast cells. We conducted TRAP staining assay by TRACP & ALP double-stain Kit (MK300; TAKARA) following the manufacturer’s protocol. Briefly, Raw264.7 cells were seeded into 48-well culture plate at density of 3 × 10^3^ cells per wells in a DMEM medium. After 24 h, DMEM medium was changed to complete differentiation alpha-MEM medium and pre-treated with different concentration of DMSO, ILG, and ILG derivatives (1, 5, and 10 μM) for 1 h. For osteoclast differentiation, we treated with 40 ng/mL of human recombinant RANKL (ALX-522-131; ENZO Life Science, Farmingdale, NY, USA) for 4 days. The fresh medium, ILG, ILG derivatives, and RANKL were retreated every 2 days. When an osteoclastogenesis was complete, the cell supernatants were removed, and cell were washed with phosphate buffered saline (PBS) one time. Total of 120 μL of fixation solution was used for fixing cells for 5 min at room temperature. Then each well was washed three times using 2 mL of sterile distilled water. Total of 120 μL of substrate solution was added to all wells to detect TRAP and the plate was covered with parafilm to prevent the sample from drying. Next, the plate was incubated at 37 °C for 45 min and then washed three times with sterile distilled water. The features of positive TRAP cells were photographed using microscope (CKX41; Olympus, Tokyo, Japan) and digital camera (eXcope T300; Olympus). The number of the positive TRAP cells that have three more nucleus was measured by NIH ImageJ 1.52a software.

### 4.5. Reverse Transcription PCR

Raw264.7 cells were seeded in 6-well culture plate at density of 3 × 10^4^ cells per wells in a DMEM medium. After cells were attached, ILG and ILG derivatives were treated to cells for 1 h before adding 40 ng/mL RANKL with α-MEM medium. Cell was incubated for 4 days and ILG, ILG derivatives, and RANKL were treated every 2 days. After 4 days of treatment, total RNAs were isolated from Raw264.7 cells using 500 μL of the TRIzol reagent (15596018; Thermo Fisher Scientific) according to general protocol. The total RNA was measured by nanodrop MD2000 spectrophotometer (Biofuture, Britain, UK). Total of 1 μg total RNA was synthesized by cDNA using the Maxime RT PreMix Kit (25081; iNtRON Biotechnology, Korea) under the manufacturer’s protocol. To compare the RNA expression, the cDNA was analyzed by reverse transcription (RT)-PCR using EmeraldAmp^®^ GT PCR Master Mix (RP310A; TAKARA. The RT-PCR was performed by PCR machine (C1000 touch Thermal cycler; Bio-Rad, Hercules, California, USA) and under the following conditions: denaturation at 95 °C for 20 sec, annealing at 56 °C for 30 s, and extension at 72 °C for 30 s for 35 cycles and, final extension at 72 °C for 5 min. All primer pairs were synthesized by Bioneer (Daejeon, Korea). The specific primers used by RT-PCR reactions were designed as follows: TRAP (forward: 5′- CTCCTGCCTGTTCTCTTCCCA-3′; reverse: 5′- AAGAGAGAAAGTCAAGGGAGT GGC-3′), Calcitonin receptor (forward: 5′- AGCTTGTTGGCACCTTTGTAT-3′; reverse: 5′-TTGC CTATGCCAGGACCAAT-3′), Cathepsin K (forward: 5′-GCAGATGTTTGTGTTGGTCTCT-3′; reverse: 5′-TGGTGGAAAGGTGTGACAGG′), DC-STAMP (forward: 5′-TTGAACCGAGCTGCATT CCT-3′; reverse: 5′-GCACTACCTTGGCCTTACCT-3′), GAPDH (forward: 5′-CAAGGTCATCCATG GACAACTTTG-3′; reverse: 5′-GTCCACCACCCTGTTGCTGTAG-3′). The GAPDH was used as the internal control for normalization.

### 4.6. Bone Resorption Assay

The bone resorbing activity of the osteoclast cells was measured by Bone resorption assay kit (CSR-BRA-48KIT; COSMOBIO, Tokyo, Japan). Raw264.7 cells were seeded onto calcium phosphate (CaP)-coated 48-well culture plate at density of 5 × 10^3^ cells per wells. Raw264.7 cells were cultured at 37 °C and 5% CO_2_ in DMEM containing 10% FBS. Subsequently, DMSO, ILG, and ILG derivatives (5 and 10 μM/mL) were added into each well for 1 h before RANKL (100 ng/mL) treatment as an inducer of osteoclast differentiation in phenol red-free α-MEM containing 10% FBS. All reagents were retreated every 2 days for 6 days. On day 6, the conditioned medium was discarded and 5% sodium hypochlorite was treated into each well for 5 min to remove the cells. After the plate was washed with D.W. and dried. The pit area, resorbed area by osteoclast cell, was captured using a digital camera (Olympus) attached to microscope (Olympus). The pit areas were measured by NIH ImageJ 1.52a software.

### 4.7. Western Blot Analysis

Western blotting was used to analyze activations of extracellular signal-regulated kinase (ERK), p38 mitogen-activated protein kinase (MAPK), c-Jun N-terminal kinase (JNK), and nuclear factor-kappaB (NF-κB) using a phospho-specific antibody. The Raw264.7 cells were seeded (5 × 10^3^ cells/well) into 6-well culture plate. After pre-treatment of ILG and ILG derivatives for 1 h, RANKL (40 ng/mL) was added into the cells for 10 min. Subsequently, the cells were washed with cold PBS for two times and harvested by RIPA lysis buffer (25 mM Tris pH 7.6, 150 mM NaCl, 1% NP-40, 1% sodium deoxycholate, 0.1% SDS) with protease inhibitor (sc-11697498001; Santa cruz) and phosphatase inhibitors (4906845001; Thermo Fisher Scientific). The lysate samples containing 20 μg of protein were separated on a 12% sodium dodecyl sulfate (SDS)-polyacrylamide gel and then transferred to immobilon-P PVDF membranes (IPVH00010; Merk Millipore, Burlington, MA, USA). Membranes were blocked with 5% skim milk in Tris-buffered saline (TBS) containing 0.1% Tween-20 (TBST) at room temperature for 1 h and incubated with primary antibodies against phospho-ERK (7383; Cell signaling), total ERK (9102; Cell signaling, Danvers, MA, USA), phospho-p38 (9211; Cell signaling), total p38 (9212; Cell signaling), phospho-JNK (9251; Cell signaling), total JNK (9252; Cell signaling), phospho-NF-kB (6956; Cell signaling), total NF-kB (3039; Cell signaling), and anti-β-actin (sc-47778; Santa cruz, Paso Robles, CA, USA). All antibodies were diluted 1:1000 in BSA except anti-β-actin (dilution ratio = 1:5000) in BSA for overnight at 4 °C. Secondary antibodies were anti-mouse (ADI-SAB-100J; ENZO) or anti-rabbit (ADI-SAB-300-J; ENZO), and used at 1:2000 dilution in 5% skim milk in TBST for 1 h at room temperature. The protein bands were visualized with an enhanced chemiluminescence method (AbClon, Seoul, Korea) and quantified by ImageJ 1.52a software. The expression levels of proteins were analyzed as the ratio of the densitometric measurement of indicated proteins in cell lysate to the corresponding internal standard (β-actin).

### 4.8. Actin Ring Formation Assay

RAW264.7 cells were seeded at a density of 1 × 10^4^ cells/well onto 4-well cell culture slide (SPL, Korea) and stimulated with RANKL (40 ng/mL) in the presence of various concentrations (5 and 10 μM) of ILG and ILG derivatives. After 4 days of incubation, the medium was gently discarded, and the cells were fixed using 4% paraformaldehyde for 10 min. The cells were washed with PBS for two times. To permeabilize the cells, PBS containing 0.1% Triton X-100 was treated into the cells for 20 min. After washing with PBS three times, cells were blocked with 1% BSA in PBS for 30 min at room temperature. The Cells were exposed to phalloidin (T7471, Invitrogen, Carlsbad, CA, USA) for 20 min at room temperature in the dark place and washed in PBS for three times. The nuclei of cells were stained with diamidino-2-phenylindole (DAPI) (D21490; Thermo Fisher Scientific) in the dark place for 3 min, and then washed tree times with PBS for 5 min. The images of F-actin ring formation were taken using a LSM 700 laser scanning confocal microscope (Carl zeiss, Overkochen, Germany).

### 4.9. Statistical Analysis

The experiments were performed at least three times. The data obtained are expressed as the mean ± standard error of the mean (SEM). Statistical analysis was performed by GraphPad Prism version 7.0 software (San Diego, CA, USA). One-way analysis of variance (ANOVA) was used to compare three or more groups followed by a Bonferroni post hoc test and Student’s *t* test was used to compare two groups. *p*-values of *p* < 0.05 were considered statistically significant.

## 5. Conclusions

In the present study, we investigated the effects of ILG and its derivatives on osteoclast differentiation and activation. Our data indicated that derivative 10, robtein, is a potent inhibitor of osteoclast differentiation and activation. This is the first report to demonstrate the anti-osteoclast differentiation effect of robtein.

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
