# Peer review of "Isoliquiritigenin Derivatives Inhibit RANKL-Induced Osteoclastogenesis by Regulating p38 and NF-κB Activation in RAW 264.7 Cells"

_molecules, 2020, doi:10.3390/molecules25173908_

Round 1

Reviewer 1 Report

The manuscript submitted by K.-C. Hwang and S. Lim described the effects of some derivatives of isoliquiritigenin, a natural product, on osteoclastogenesis. More specifically, the reported compounds aim to regulate p38 and NF-kappaB activations, thus inhibiting RANKL-induced osteoclastogenesis.

This work is well-conducted and, regarding to the findings included in this manuscript, could be published in Molecules after some major modifications. Some clarity on the obtained results must be brought as the readers of Molecules are not all specialized in cell biology.

Authors have to mention what are the anti-osteoclastic drugs designed to suppress excessive osteoclast activation and what type of side-effects are related, together with literature references (only one paper in reference 12 is not enough).

A figure with the reported anti-osteoclastic drugs, with a special emphasis on the molecules targeting the same mechanism would be appreciated.

As there is no full stop at the end and as it ends as “newly synthesized ILG derivatives were screened”, I guess the abstract is incomplete as no indication was provided on biological activities of this derivatives… Isn’t it?

I would strongly the authors to add some more literature references dealing with anti-osteoclastic effects of isoliquiritigenin as there are previous papers on the subject (please give more than one reference in footnote 13). The interest of this work is to describe the effects of derivatives of this natural product but authors have to mention the literature precedence of such effect, especially if it is related to RANKL. If there are some proofs of reduced side effects with ILG derivatives, this should be emphasized. I also strongly suggest the authors to add a short part of the manuscript on the potential estrogenic effects of these structures, with literature references.

At the end of the introduction, please modify the sentence “were newly synthesized and screened” as derivatives 1-12 are not new… Only the screening of the activity was not described.

One of the major remark I can formulate is that there is no general Scheme for the preparation of compounds 1-12.

I agree that authors have cited the references dealing with the preparation of such derivatives (according to literature references) but it would be informative for the reader to have a synthetic scheme (with reference numbers eg under the arrows). So please provide such a scheme.

Personally, I do not see the interest to add in Table 1 the IUPAC names of derivatives 1-12 as these names are included in the supplementary data.

All along the text, authors indicate “#3 #10” to name the evaluated compounds but this notation is not homogenous with Table 1 or the following figures. So please remove “#”.

Another major remark is that only the activity of ILG, compounds 3 and 10 were described. What is the activity of all the other compounds of Table 1. Even though these derivatives are not active, this absence of activity must be described in the paper. If the authors do not want to describe this important part in the main manuscript, they must include these results in the supplementary section. At least, a quantification of the activity of each compound could be included in Table 1.

This is an important remark of the reviewer.

In paragraph 2.1.3 (and associated figures), authors mentioned a synergistic effect between 3 and 10. This is a very interesting finding, that must be explained more deeply. Please provide some hypothesis, especially in term of structural diversity (3 is a chromane derivative whereas 10 is a chalcone). These two compounds might use in synergy but please provide possible explanation.

Authors must mention the origin of RAW264.7 cells in the text: from which organ are these cells originated? This is explained is paragraph 4.1 but please add this precision in the text.

In all the Figures, authors have to give appropriate legends/notes. I am sorry to mention that but this is not clear at all… especially for the readers of Molecules as they are not in all cases specialists in cell biology.

For example, what is “com”? I guess it is the combination of compounds 3 and 10 but this MUST be clearly explained.

So, please consider this mandatory modification, especially for figures 1, 2 and 3 (explanations are more detailed for the other figures).

For the non-specialist reader, could you please define what “pit area” is?

I know that some abbreviations are well described in the literature but, when used for the first time in the paper (WST, LDH…) they must be clearly defined in the text.

In the concluding part of the discussion, I suggest the authors not to focus on the in vivo studies because some pharmacomodulation of Robtein are still possible and this should appear in the manuscript (at the end of the discussion and in the conclusion).

In the discussion, some indications on the interaction mechanism of most active compounds on its plausible target have to be brought. I know that Molecules is not a medicinal chemistry journal (though it is mentioned in its scope) but this should be added.

A correction must be done in the second line of the conclusion “robtain”.

All the journal abbreviations of the bibliographic references must be carefully checked using the CASSI search tool. Please find some errors that must be corrected (not an exhaustive list…): Bone Res. (with a dot…), journal name for ref. 2 appears unknown to me… is it “Front. Med., [Proc. Symp.]” (if yes, please use the correct name…)? Same thing for ref. 3 “Immune Netw”? “J. Clin. Invest.” “EMBO J.” With dots… So please check evertything carefully.

In the supporting information, some 1H and 13C NMR spectra are of bad quality. For example, in the spectrum of isoliquiritigenin, the y-scale was set to residual DMSO solvent, so that the peaks of the compound appear too small… Please set the y-scale of the spectrum to see more details of the compound peaks. Same remark for compounds 1, 3, 4, 5, 9, 10, 11 and 12.

Similarly, signal/noise ratios for several compounds are not satisfactory: ILG, derivatives 1, 2, 4, 9, 11 + please correct the y-scale.

Author Response

Dear Editor,

We authors very much appreciated the encouraging, critical and constructive comments and suggestions on this manuscript by the reviewers. The comments have been very thorough and useful in improving the manuscript. We strongly believe that the comments and suggestions have significantly increased the scientific value of revised manuscript. We are submitting the corrected manuscript with consolidated data. The manuscript has been revised as per the comments given by the reviewer, and our responses to all the comments are as follows:

Reviewer 1

Comments and Suggestions for Authors

The manuscript submitted by K.-C. Hwang and S. Lim described the effects of some derivatives of isoliquiritigenin, a natural product, on osteoclastogenesis. More specifically, the reported compounds aim to regulate p38 and NF-kappaB activations, thus inhibiting RANKL-induced osteoclastogenesis. This work is well-conducted and, regarding to the findings included in this manuscript, could be published in Molecules after some major modifications. Some clarity on the obtained results must be brought as the readers of Molecules are not all specialized in cell biology.

-Authors have to mention what are the anti-osteoclastic drugs designed to suppress excessive osteoclast activation and what type of side-effects are related, together with literature references (only one paper in reference 12 is not enough). A figure with the reported anti-osteoclastic drugs, with a special emphasis on the molecules targeting the same mechanism would be appreciated.

Response: More references with anti-osteoclastic drugs and related side effects were added in Introduction. As for the second request, we are afraid that we clearly understand the recommendation given by the reviewer. We are not sure whether we need to draw schematic figure for our new findings or something. Please let us know, then we will try our best to accommodate the reviewer’s suggestions.

- As there is no full stop at the end and as it ends as “newly synthesized ILG derivatives were screened”, I guess the abstract is incomplete as no indication was provided on biological activities of this derivatives… Isn’t it?

Response: We assume the last part of the abstract had been accidentally cut off during the uploading process. Sorry for the inconvenience. The last part of the abstract has been correctly incorporated in the revised manuscript.

-I would strongly the authors to add some more literature references dealing with anti-osteoclastic effects of isoliquiritigenin as there are previous papers on the subject (please give more than one reference in footnote 13).

Response: A number of new references concerning the anti-osteoclastic effects of isoliquiritigenin has been added as the reviewer suggested.

-The interest of this work is to describe the effects of derivatives of this natural product but authors have to mention the literature precedence of such effect, especially if it is related to RANKL. If there are some proofs of reduced side effects with ILG derivatives, this should be emphasized. I also strongly suggest the authors to add a short part of the manuscript on the potential estrogenic effects of these structures, with literature references.

Response: We added the possible estrogenic effect of ILG and its chalcone derivatives in Discussion section (last paragraph) as the reviewer suggested.

- At the end of the introduction, please modify the sentence “were newly synthesized and screened” as derivatives 1-12 are not new… Only the screening of the activity was not described.

Response: “newly” is removed.

-One of the major remark I can formulate is that there is no general Scheme for the preparation of compounds 1-12. I agree that authors have cited the references dealing with the preparation of such derivatives (according to literature references) but it would be informative for the reader to have a synthetic scheme (with reference numbers eg under the arrows). So please provide such a scheme.

Response: We added the synthetic scheme for each of compounds which is in Supplementary data as Scheme 1.

- Personally, I do not see the interest to add in Table 1 the IUPAC names of derivatives 1-12 as these names are included in the supplementary data.

Response: Table 1 has been moved to the supplementary data all together.

-All along the text, authors indicate “#3 #10” to name the evaluated compounds but this notation is not homogenous with Table 1 or the following figures. So please remove “#”.

Response: We removed “#” from the manuscript.

-Another major remark is that only the activity of ILG, compounds 3 and 10 were described. What is the activity of all the other compounds of Table 1. Even though these derivatives are not active, this absence of activity must be described in the paper. If the authors do not want to describe this important part in the main manuscript, they must include these results in the supplementary section. At least, a quantification of the activity of each compound could be included in Table 1. This is an important remark of the reviewer.

Response: We added results of TRAP staining (1-13) as Figure S13.

-In paragraph 2.1.3 (and associated figures), authors mentioned a synergistic effect between 3 and 10. This is a very interesting finding, that must be explained more deeply. Please provide some hypothesis, especially in term of structural diversity (3 is a chromane derivative whereas 10 is a chalcone). These two compounds might use in synergy but please provide possible explanation.

Response: There was a mistake and now the number is fixed (ILG is no.1 and derivatives are no.2-13). So both 3 and 10 are chalcone. Therefore we added the protective effect of chalcone structure in Discussion section.

- Authors must mention the origin of RAW264.7 cells in the text: from which organ are these cells originated? This is explained is paragraph 4.1 but please add this precision in the text.

Response: We added the detail of raw264.7 in the Result section (2.1.1.)

-In all the Figures, authors have to give appropriate legends/notes. I am sorry to mention that but this is not clear at all… especially for the readers of Molecules as they are not in all cases specialists in cell biology. For example, what is “com”? I guess it is the combination of compounds 3 and 10 but this MUST be clearly explained. So, please consider this mandatory modification, especially for figures 1, 2 and 3 (explanations are more detailed for the other figures).

Response: Mandatory modification was done for all figures.

-For the non-specialist reader, could you please define what “pit area” is?

Response: Detailed explanation was added in Results (2.1.2.)

-I know that some abbreviations are well described in the literature but, when used for the first time in the paper (WST, LDH…) they must be clearly defined in the text.

Response: I carefully read through the manuscript in which the abbreviations were added.

-In the concluding part of the discussion, I suggest the authors not to focus on the in vivo studies because some pharmacomodulation of Robtein are still possible and this should appear in the manuscript (at the end of the discussion and in the conclusion). In the discussion, some indications on the interaction mechanism of most active compounds on its plausible target have to be brought. I know that Molecules is not a medicinal chemistry journal (though it is mentioned in its scope) but this should be added.

Response: We removed descriptions on in vivo studies from the discussion and conclusion. We also agree with your opinion for the interaction mechanism between robtein and plausible target. However, Intra and extracellular binding targets for Robtein are hardly expected as there are almost no previous studies using Robtein. Instead, we described the inhibitory action for the common regulatory mechanism in osteoclast activation such as P38 and NF-kB in the first paragraph of Discussion.

-A correction must be done in the second line of the conclusion “robtain”.

Response: We corrected.

-All the journal abbreviations of the bibliographic references must be carefully checked using the CASSI search tool. Please find some errors that must be corrected (not an exhaustive list…): Bone Res. (with a dot…), journal name for ref. 2 appears unknown to me… is it “Front. Med., [Proc. Symp.]” (if yes, please use the correct name…)? Same thing for ref. 3 “Immune Netw”? “J. Clin. Invest.” “EMBO J.” With dots… So please check evertything carefully.

Response: References were carefully checked and fixed.

-In the supporting information, some 1H and 13C NMR spectra are of bad quality. For example, in the spectrum of isoliquiritigenin, the y-scale was set to residual DMSO solvent, so that the peaks of the compound appear too small… Please set the y-scale of the spectrum to see more details of the compound peaks. Same remark for compounds 1, 3, 4, 5, 9, 10, 11 and 12. Similarly, signal/noise ratios for several compounds are not satisfactory: ILG, derivatives 1, 2, 4, 9, 11 + please correct the y-scale.

Response: Unlike compounds used for experiments (large scale synthesis), some of compounds were synthesized additionally as small amount for this analysis, as we used almost all of them for in vitro experiments. For example, isoliquiritigenin (no.1) and 2 were synthesized again for this analysis, so then peaks appeared too small. We corrected the y-scale and signals for other compounds. Modified results are in the Supplementary Figures 1-12. Because of due date, we corrected and modulated the peak for this revision. However, if you insist that we need to do more experiments, we are willing to synthesize the compounds again and analyze. Please let us know.

Reviewer 2 Report

This study investigated the effects of Isoliquiritigenin (ILG) and its derivatives on osteoclastogenesis using RAW264.7 cells. The authors prepared 12 derivatives of ILG and treated RAW264.7 cells with them prior to RANKL-induced osteoclast differentiation. The results of tartrate-resistant acid phosphatase (TRAP) staining revealed that the two derivatives of ILG, No. 3 and No. 10, significantly inhibited the differentiation of RAW264.7 cells into osteoclast-like cells without cytotoxicity. These derivatives also inhibited that osteoclast specific genes, such as TRAP cathepsin K, DC-STAMP, calcitonin receptor. Pit assay and immunofluorescence staining indicated that derivative No.3 and derivative No. 10 suppressed the dissolution of the inorganic component and the actin formation that is important for bone resorption. The authors also investigated that the combination effects of derivative No. 3 and No. 10 in osteoclastogenesis; the number of TRAP positive cells, the dissolution of the inorganic component and the phosphorylation of p38 and NF-kB, which was signaling pathway involved in osteoclast differentiation, were decreased in the presence of derivative No. 3 and No. 10 compared to the presence of either a No. 3 and No. 10. Based on these results, they concluded these two ILG derivatives could be a candidate of anti-osteoclastic drug. 

This manuscript includes interesting topics, the story of it is straightforward, and the experimental evidences needed to the conclusions are mostly convincing. I would like to recommend to mention about the effects of LIG or ILG derivatives in the expression of carbonic anhydrase II and/or V-ATPase d2. This is because the authors indicated the data of bone resorption assay using calcium phosphate-coated plate, which did not contain organic component. Carbonic anhydrase II and V-ATPase d2 are essential factors to increase the level of H+ in bone resorptive microenvironment surrounded by actin rings. Rather than cathepsin K and TRAP, carbonic anhydrase II and V-ATPase d2 might be able to explain the mechanism of suppressive effects of ILG and its derivatives in dissolution of inorganic phase of bone (Figure 4 B, C, Figure 5D).

Author Response

We authors very much appreciated the encouraging, critical and constructive comments and suggestions on this manuscript by the reviewers. The comments have been very thorough and useful in improving the manuscript. We strongly believe that the comments and suggestions have significantly increased the scientific value of revised manuscript. We are submitting the corrected manuscript with consolidated data. The manuscript has been revised as per the comments given by the reviewer, and our responses to all the comments are as follows:

Comments and Suggestions for Authors

This study investigated the effects of Isoliquiritigenin (ILG) and its derivatives on osteoclastogenesis using RAW264.7 cells. The authors prepared 12 derivatives of ILG and treated RAW264.7 cells with them prior to RANKL-induced osteoclast differentiation. The results of tartrate-resistant acid phosphatase (TRAP) staining revealed that the two derivatives of ILG, No. 3 and No. 10, significantly inhibited the differentiation of RAW264.7 cells into osteoclast-like cells without cytotoxicity. These derivatives also inhibited that osteoclast specific genes, such as TRAP cathepsin K, DC-STAMP, calcitonin receptor. Pit assay and immunofluorescence staining indicated that derivative No.3 and derivative No. 10 suppressed the dissolution of the inorganic component and the actin formation that is important for bone resorption. The authors also investigated that the combination effects of derivative No. 3 and No. 10 in osteoclastogenesis; the number of TRAP positive cells, the dissolution of the inorganic component and the phosphorylation of p38 and NF-kB, which was signaling pathway involved in osteoclast differentiation, were decreased in the presence of derivative No. 3 and No. 10 compared to the presence of either a No. 3 and No. 10. Based on these results, they concluded these two ILG derivatives could be a candidate of anti-osteoclastic drug. 

This manuscript includes interesting topics, the story of it is straightforward, and the experimental evidences needed to the conclusions are mostly convincing.

I would like to recommend to mention about the effects of ILG or ILG derivatives in the expression of carbonic anhydrase II and/or V-ATPase d2. This is because the authors indicated the data of bone resorption assay using calcium phosphate-coated plate, which did not contain organic component. Carbonic anhydrase II and V-ATPase d2 are essential factors to increase the level of H+ in bone resorptive microenvironment surrounded by actin rings. Rather than cathepsin K and TRAP, carbonic anhydrase II and V-ATPase d2 might be able to explain the mechanism of suppressive effects of ILG and its derivatives in dissolution of inorganic phase of bone (Figure 4 B, C, Figure 5D).

Response: We added the discussion based on reviewer’s comment (blue highlight)

Round 2

Reviewer 1 Report

I have read the revised version of the manuscript submitted by by K.-C. Hwang and S. Lim, which described the effects of some derivatives of isoliquiritigenin on osteoclastogenesis.

The authors have modified their manuscript, according to most of my recommendations.

Hence, I recommend the publication of this work in Molecules, pending some minor modifications.

Indeed, two of my major remarks were not taken in consideration. So, please modify your revised manuscript accordingly:

- provide a GENERAL scheme for the preparation of isoliquiritigenin derivatives (that were evaluated in this study) in the main text. I have noticed that authors have completed the supplementary information with detailed steps of the preparation of all compounds studied in this paper (please give the yields for each step in the SI) but I asked for a general scheme in the main text.

- provide a figure in which the chemical structures of known (ie already reported in the literature) anti-osteoclastic drugs (the compounds reported in the added references) are depicted.

These two remarks and important for me and this must be corrected in the revised manuscript, as I suggested. Molecules is a journal dedicated to molecular entities and their biological activities. So, the structure and synthetic strategies to reach those entities, and compounds already described in the literature must be described.

Moreover, remove “novel” in the last sentence of the abstract are these compounds were already described in the literature.

For NMR spectra in the SI, as these compounds were already described in the literature, there is no need to provide higher quality spectra.

Author Response

Dear. Reviewer

We authors very much appreciated the encouraging, critical and constructive comments and suggestions on this manuscript by the reviewer. The comments have been very thorough and useful in improving the manuscript. Minor revised parts are highlighted in pink color for your convenience of re-reviewing.

Comments and Suggestions for Authors

I have read the revised version of the manuscript submitted by by K.-C. Hwang and S. Lim, which described the effects of some derivatives of isoliquiritigenin on osteoclastogenesis.

The authors have modified their manuscript, according to most of my recommendations.

Hence, I recommend the publication of this work in Molecules, pending some minor modifications. Indeed, two of my major remarks were not taken in consideration. So, please modify your revised manuscript accordingly:

- provide a GENERAL scheme for the preparation of isoliquiritigenin derivatives (that were evaluated in this study) in the main text. I have noticed that authors have completed the supplementary information with detailed steps of the preparation of all compounds studied in this paper (please give the yields for each step in the SI) but I asked for a general scheme in the main text.

Response: We added general scheme as 4.1. Synthesis in Materials and Methods section and yields for each step was added Supplementary results which are highlighted as pink color.

- provide a figure in which the chemical structures of known (ie already reported in the literature) anti-osteoclastic drugs (the compounds reported in the added references) are depicted.

Response: We added the chemical structures as Figure 1.

These two remarks and important for me and this must be corrected in the revised manuscript, as I suggested. Molecules is a journal dedicated to molecular entities and their biological activities. So, the structure and synthetic strategies to reach those entities, and compounds already described in the literature must be described.

Moreover, remove “novel” in the last sentence of the abstract are these compounds were already described in the literature.

Response: We removed “novel” form the abstract. Thank you very much for careful review.

For NMR spectra in the SI, as these compounds were already described in the literature, there is no need to provide higher quality spectra.

Response: I appreciate your understanding.